# Neural Point Process for Learning Spatiotemporal Event Dynamics

**Zihao Zhou**                                            ZIZ244@UCSD.EDU
*UC San Diego*

**Xingyi Yang**                                           XYANG@U.NUS.EDU
*National University of Singapore*

**Ryan Rossi**                                            RYROSSI@ADOBE.COM
*Adobe Research*

**Handong Zhao**                                          HAZHAO@ADOBE.COM
*Adobe Research*

**Rose Yu**                                               ROSEYU@UCSD.EDU
*UC San Diego*

## Abstract

Learning the dynamics of spatiotemporal events is a fundamental problem. Neural point processes enhance the expressivity of point process models with deep neural networks. However, most existing methods only consider temporal dynamics without spatial modeling. We propose Deep Spatiotemporal Point Process (`DeepSTPP`), a deep dynamics model that integrates spatiotemporal point processes. Our method is flexible, efficient, and can accurately forecast irregularly sampled events over space and time. The key construction of our approach is the nonparametric space-time intensity function, governed by a latent process. The intensity function enjoys closed form integration for the density. The latent process captures the uncertainty of the event sequence. We use amortized variational inference to infer the latent process with deep networks. Using synthetic datasets, we validate our model can accurately learn the true intensity function. On real-world benchmark datasets, our model demonstrates superior performance over state-of-the-art baselines.

**Keywords:** spatiotemporal dynamics, neural point processes, kernel density estimation

## 1. Introduction

Accurate modeling of spatiotemporal event dynamics is fundamentally important for disaster response (Veen and Schoenberg, 2008), logistic optimization (Safikhani et al., 2018) and social media analysis (Liang et al., 2019). Compared to other sequence data such as texts or time series, spatiotemporal events occur irregularly with uneven time and space intervals.

Discrete-time deep dynamics models such as recurrent neural networks (RNNs) (Hochreiter and Schmidhuber, 1997; Chung et al., 2014) assume events to be evenly sampled. Interpolating an irregular sampled sequence into a regular sequence can introduce significant biases (Rehfeld et al., 2011). Furthermore, event sequences contain strong spatiotemporal dependencies. The rate of an event depends on the preceding events, as well as the events geographically correlated to it.

Spatiotemporal point processes (STPP) (Daley and Vere-Jones, 2007; Reinhart et al., 2018) provides the statistical framework for modeling continuous-time event dynamics. As shown in Figure 1, given the history of events sequence, STPP estimates the intensity function that is evolv-

ing in space and time. However, traditional statistical methods for estimating STPPs often require strong modeling assumptions, feature engineering, and can be computationally expensive.

Machine learning community is observing a growing interest in continuous-time deep dynamics models that can handle irregular time intervals. For example, Neural ODE (Chen et al., 2018) parametrizes the hidden states in an RNN with an ODE. Shukla and Marlin (2018) uses a separate network to interpolates between reference time points. Neural temporal point process (TPP) (Mei and Eisner, 2017; Zhang et al., 2020; Zuo et al., 2020) is an exciting area that combines fundamental concepts from temporal point processes with deep learning to model continuous-time event sequences, see a recent review on neural TPP (Shchur et al., 2021). However, most of the existing models only focus on *temporal* dynamics without considering *spatial* modeling.

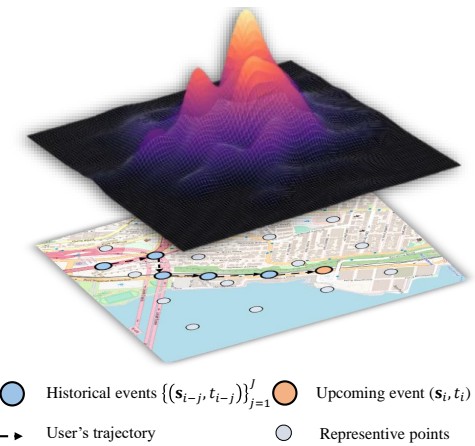

Historical events $\{(\mathbf{s}_{i-j}, t_{i-j})\}_{j=1}^{J}$    Upcoming event $(\mathbf{s}_i, t_i)$

User's trajectory    Representive points

Figure 1: Illustration of learning spatiotemporal point process. We aim to learn the space-time intensity function given the historical event sequence and representative points as background.

In the real world, while time is a unidirectional process (arrow of time), space extends in multiple directions. This fundamental difference from TPP makes it nontrivial to design a unified STPP model. The naive approach to approximate the intensity function by a deep neural network would lead to intractable integral computation for likelihood. Prior research such as Du et al. (2016) discretizes the space as "markers" and use marked TPP to classify the events. This approach cannot produce the space-time intensity function. Okawa et al. (2019) models the spatiotemporal density using a mixture of symmetric kernels, which ignores the unidirectional property of time. Chen et al. (2021) proposes to model temporal intensity and spatial density separately with neural ODE, which is computational expensive.

We propose a simple yet efficient approach to learn STPP. Our model, *Deep Spatiotemporal Point Process* (`DeepSTPP`) marries the principles of spatiotemporal point processes with deep learning. We take a non-parametric approach and model the space-time intensity function as mixture of kernels. The parameters of the intensity function are governed by a latent stochastic process no sampling which captures the uncertainty of the event sequence. The latent process is then inferred via amortized variational inference. That is, we draw a sample from the variational distribution for every event. We use a Transformer network to parametrize the variational distribution conditioned on the previous events.

Compared with existing approaches, our model is non-parametric, hence does not make assumptions on the parametric form of the distribution. Our approach learns the space-time intensity function jointly without requiring separate models for time-intensity function and spatial density as in Chen et al. (2021). Our model is probabilistic by nature and can describe various uncertainties in the data. More importantly, our model enjoys closed form integration, making it feasible for processing large-scale event datasets. To summarize, our work makes the following key contributions:

- **Deep Spatiotemporal Point Process.** We propose a novel Deep Point Process model for forecasting unevenly sampled spatiotemporal events. It integrates deep learning with spatiotemporal point processes to learn continuous space-time dynamics.

- **Neural Latent Process.** We model the space-time intensity function using a nonparametric approach, governed by a latent stochastic process. We use amortized variational inference to perform inference on the latent process conditioned on the previous events.

- **Effectiveness.** We demonstrate our model using many synthetic and real-world spatiotemporal event forecasting tasks, where it achieves superior performance in accuracy and efficiency. We also derive and implement efficient algorithms for simulating STPPs.

## 2. Methodology

We first introduce the background of spatiotemporal point process, and then describe our approach to learn the underlying spatiotemporal event dynamics.

### 2.1. Background on Spatiotemporal Point Process

**Spatiotemporal Point Process.** Spatiotemporal point process (STPP) models the number of events $N(\mathcal{S} \times (a, b))$ that occurred in the Cartesian product of the spatial domain $\mathcal{S} \subseteq \mathbb{R}^2$ and the time interval $(a, b]$. It is characterized by a non-negative *space-time intensity function* given the history $\mathcal{H}_t := \{(\mathbf{s}_1, t_1), \ldots, (\mathbf{s}_n, t_n)\}_{t_n \leq t}$:

$$\lambda^*(\mathbf{s}, t) := \lim_{\Delta \mathbf{s} \to 0, \Delta t \to 0} \frac{\mathbb{E}[N(B(\mathbf{s}, \Delta \mathbf{s}) \times (t, t + \Delta t))|\mathcal{H}_t]}{B(\mathbf{s}, \Delta \mathbf{s})\Delta t} \tag{1}$$

which is the probability of finding an event in an infinitesimal time interval $(t, t + \Delta t]$ and an infinitesimal spatial ball $\mathcal{S} = B(\mathbf{s}, \Delta \mathbf{s})$ centered at location $\mathbf{s}$.

*Example 1: Spatiotemporal Hawkes process (STH).* Spatiotemporal Hawkes (or self-exciting) process assumes every past event has an additive, positive, decaying, and spatially local influence over future events. Such a pattern resembles neuronal firing and earthquakes. It is characterized by the following intensity function (Reinhart et al., 2018):

$$\lambda^*(\mathbf{s}, t) := \mu g_0(\mathbf{s}) + \sum_{i:t_i < t} g_1(t, t_i)g_2(\mathbf{s}, \mathbf{s}_i) : \mu > 0 \tag{2}$$

where $g_0(\mathbf{s})$ is the probability density of a distribution over $\mathcal{S}$, $g_1$ is the triggering kernel and is often implemented as the exponential decay function, $g_1(\Delta t) := \alpha \exp(-\beta \Delta t) : \alpha, \beta > 0$, and $g_2(\mathbf{s}, \mathbf{s}_i)$ is the density of an unimodal distribution over $\mathcal{S}$ centered at $\mathbf{s}_i$.

*Example 2: Spatiotemporal Self-Correcting process (STSC).* Self-correcting spatiotemporal point process Isham and Westcott (1979) assumes that the background intensity increases with a varying speed at different locations, and the arrival of each event reduces the intensity nearby. STSC can model certain regular event sequences, such as an alternating home-to-work travel sequence. It has the following intensity function:

$$\lambda^*(\mathbf{s}, t) = \mu \exp \left( g_0(\mathbf{s})\beta t - \sum_{i:t_i < t} \alpha g_2(\mathbf{s}, \mathbf{s}_i) \right) : \alpha, \beta, \mu > 0 \tag{3}$$

Here $g_0(\mathbf{s})$ is the density of a distribution over $\mathcal{S}$, and $g_2(\mathbf{s}, \mathbf{s}_i)$ is the density of an unimodal distribution over $\mathcal{S}$ centered at location $\mathbf{s}_i$.

**Maximum likelihood Estimation.** Given a history of $n$ events $\mathcal{H}_t$, the joint log-likelihood function of the observed events for STPP is as follows:

$$\log p(\mathcal{H}_t) = \sum_{i=1}^{n} \log \lambda^*(\mathbf{s}_i, t_i) - \int_{\mathcal{S}} \int_0^t \lambda^*(\mathbf{u}, \tau) d\mathbf{u} d\tau \tag{4}$$

Here, the space-time intensity function $\lambda^*(\mathbf{s}, t)$ plays a central role. Maximum likelihood estimation seeks the optimal $\lambda^*(\mathbf{s}, t)$ from data that optimizes Eqn. (4).

**Predictive distribution.** Denote the probability density function (PDF) for STPP as $f(\mathbf{s}, t | \mathcal{H}_t)$ which represents the conditional probability that next event will occur at location $\mathbf{s}$ and time $t$, given the history. The PDF is closely related to the intensity function:

$$f(\mathbf{s}, t | \mathcal{H}_t) \quad = \frac{\lambda^*(\mathbf{s}, t)}{1 - F^*(\mathbf{s}, t | \mathcal{H}_t)} = \lambda^*(\mathbf{s}, t) \exp\left(- \int_{\mathcal{S}} \int_{t_n}^t \lambda^*(\mathbf{u}, \tau) d\tau d\mathbf{u}\right) \tag{5}$$

where $F$ is the cumulative distribution function (CDF), see derivations in Appendix A.1. This means the intensity function specifies the expected number of events in a region conditional on the past.

The predicted time of the next event is the expected value of the predictive distribution for time $f^\star(t)$ in the entire spatial domain:

$$\mathbb{E}[t_{n+1} | \mathcal{H}_t] = \int_{t_n}^{\infty} t \int_{\mathcal{S}} f^*(\mathbf{s}, t) d\mathbf{s} dt = \int_{t_n}^{\infty} t \exp\left(- \int_{t_n}^t \lambda^*(\tau) d\tau\right) \lambda^*(t) d\mathbf{s} dt$$

Similarly, the predicted location of the next event evaluates to:

$$\mathbb{E}[\mathbf{s}_{n+1} | \mathcal{H}_t] = \int_{\mathcal{S}} \mathbf{s} \int_{t_n}^{\infty} f^*(\mathbf{s}, t) dt d\mathbf{s} = \int_{t_n}^{\infty} \exp\left(- \int_{t_n}^t \lambda^*(\tau) d\tau\right) \int_{\mathcal{S}} \mathbf{s} \lambda^*(\mathbf{s}, t) d\mathbf{s} dt$$

Unfortunately, Eqn. (4) is generally intractable. It requires either strong modeling assumptions or expensive Monte Carlo sampling. We propose the Deep STPP model to simplify the learning.

## 2.2. Deep Spatiotemporal Point Process (DSTPP)

We propose `DeepSTPP`, a simple and efficient approach for learning the space-time event dynamics. Our model (1) introduces a latent process to capture the uncertainty (2) parametrizes the latent process with deep neural networks to increase model expressivity and (3) approximates the intensity function with a set of spatial and temporal kernel functions.

**Neural latent process.** Given a sequence of $n$ event, we wish to model the conditional density of observing the next event given the history $f(\mathbf{s}, t | \mathcal{H}_t)$. We introduce a latent process to capture the uncertainty of the event history and infer the latent process with armotized variational inference. The latent process dictates the parameters in the space-time intensity function. We sample from the latent process using the re-parameterization trick Kingma and Welling (2013).

As shown in Figure 2, given the event sequence $\mathcal{H}_t = \{(\mathbf{s}_1, t_1), \dots, (\mathbf{s}_n, t_n)\}_{t_n \leq t}$, we encode the entire sequence into the high-dimensional embedding. We use positional encoding to encode the sequence order. To capture the stochasticity in the temporal dynamics, we introduce a latent process

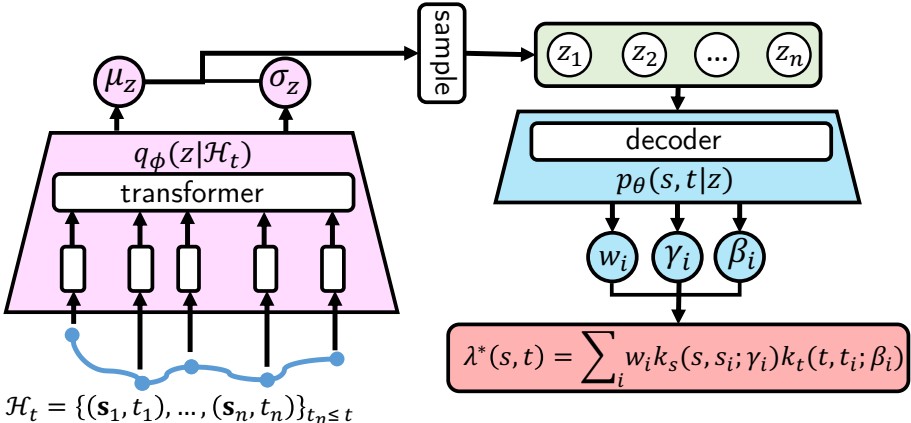

Figure 2: Design of our `DeepSTPP` model. For a historical event sequence, we encode it with a transformer network and map to the latent process $(z_1, \cdots, z_n)$. We use a decoder to generate the parameters $(w_i, \gamma_i, \beta_i)$ for each event $i$ given the latent process. The estimate intensity is calculated using kernel functions $k_s$ and $k_t$ and the decoded parameters.

$z = (z_1, \cdots, z_n)$ for the entire sequence. We assume the latent process follows a multivariate Gaussian at each time step:

$$z_i \sim q_\phi(z_i|\mathcal{H}_t) = \mathcal{N}(\mu, \mathrm{Diag}(\sigma)) \tag{6}$$

where the mean $\mu$ and covariance $\mathrm{Diag}(\sigma)$ are the outputs of the embedding neural network. In our implementation, we found using a Transformer Vaswani et al. (2017) with sinusoidal positional encoding to be beneficial. The positions to be encoded are the normalized event time instead of the index number, to account for the unequal time interval. Recently, Zuo et al. (2020) also demonstrated that Transformer enjoys better performance for learning the intensity in temporal point processes.

**Non-parametric model.** We take a non-parameteric approach to model the space-time intensity function $\lambda^*(\mathbf{s}, t)$ as:

$$\lambda^*(\mathbf{s}, t|z) = \sum_{i=1}^{n+J} w_i k_s(\mathbf{s}, \mathbf{s}_i; \gamma_i) k_t(t, t_i; \beta_i) \tag{7}$$

Here $w_i(z), \gamma_i(z), \beta_i(z)$ are the parameters for each event that is conditioned on the latent process. Specifically, $w_i$ represents the non-negative intensity magnitude, implemented with a soft-plus activation function. $k_s(\cdot, \cdot)$ and $k_t(\cdot, \cdot)$ are the spatial and temporal kernel functions, respectively. For both kernel functions, we parametrize them as a normalized RBF kernel:

$$k_s(\mathbf{s}, \mathbf{s}_i) = \alpha^{-1} \exp\left(-\gamma_i \|\mathbf{s} - \mathbf{s}_i\|\right), \quad k_t(t, t_i) = \exp\left(-\beta_i \|t - t_i\|\right) \tag{8}$$

where the bandwidth parameter $\gamma_i$ controls an event's influence over the spatial domain. The parameter $\beta_i$ is the decay rate that represents the event's influence over time. $\alpha = \int_{\mathcal{S}} \exp\left(-\gamma_i \|\mathbf{s} - \mathbf{s}_i\|\right) d\mathbf{s}$ is the normalization constant.

We use a decoder network to generate the parameters $\{w_i, \gamma_i, \beta_i\}$ given $z$ separately, shown in Figure 2. Each decoder is a 4-layer feed-forward network. We use a softplus activation function to

ensure $w_i$ and $\gamma_i$ are positive. The decay rate $\beta_i$ can be any number, such that an event could have constant or increasing triggering intensity over time.

In addition to $n$ historical events, we also randomly sample $J$ representative points from the spatial domain to approximate the background intensity. This is to account for the influence from unobserved events in the background, with varying rates at different absolution locations. The inclusion of these representative points can approximate this background distribution.

The model design in (7) enjoys a closed form integration, which gives the conditional PDF as:

$$f(\mathbf{s}, t|\mathcal{H}_t, z) = \lambda^*(\mathbf{s}, t|z) \exp\left(-\sum_{i=1}^{n+J} \frac{w_i}{\beta_i}[k_t(t_n, t_i) - k_t(t, t_i)]\right) \tag{9}$$

See the derivation details in Appendix A.2. `DeepSTPP` circumvents the integration of the intensity function and enjoys fast inference in forecasting future events. In contrast, NSTPP Chen et al. (2021) is relatively inefficient as its ODE solver also requires additional numerical integration.

**Parameter learning.** Due to the latent process, the posterior becomes intractable. Instead, we use amortized inference by optimizing the evidence lower bound (ELBO) of the likelihood. In particular, given event history $\mathcal{H}_t$, the conditional log-likelihood of the next event is:

$$\log p(\mathbf{s}, t|\mathcal{H}_t) \geq \log p_\theta(\mathbf{s}, t|\mathcal{H}_t, z) + \text{KL}(q_\phi(z|\mathcal{H}_t)||p(z)) \tag{10}$$

$$= \log \lambda^*(\mathbf{s}, t|z) - \int_{t_n}^t \lambda^*(\tau)d\tau + \text{KL}(q||p) \tag{11}$$

where $\phi$ represents the parameters of the encoder network and $\theta$ are the parameters of the decoder network. $p(z)$ is the prior distribution, which we assume to be Gaussian. $\text{KL}(\cdot||\cdot)$ is the Kullback–Leibler divergence between two distributions. We can optimize the objective function in Eqn. (11) w.r.t. the parameters $\phi$ and $\theta$ using back-propagation.

## 3. Related Work

**Spatiotemporal Dynamics Learning.** Modeling the spatiotemporal dynamics of a system in order to forecast the future is a fundamental task in many fields. Most work on spatiotemporal dynamics has been focused on spatiotemporal data measured at regular space-time interval, e.g., (Xingjian et al., 2015; Li et al., 2018; Yao et al., 2019; Fang et al., 2019; Geng et al., 2019). For discrete spatiotemporal events, statistical methods include space-time point process, see (Moller and Waagepetersen, 2003; Mohler et al., 2011). (Zhao et al., 2015) propose multi-task feature learning whereas (Yang et al., 2018) propose RNN-based model to predict spatiotemporal check-in events. These discrete-time models assume data are sampled evenly, thus are unsuitable for our task.

**Continous Time Sequence Models.** Continuous time sequence models provide an elegant approach for describing irregular sampled time series. For example, (Chen et al., 2018; Jia and Benson, 2019; Dupont et al., 2019; Gholami et al., 2019; Finlay et al., 2020; Kidger et al., 2020; Norcliffe et al., 2021) assumes the latent dynamics are continuous and can be modeled by an ODE. But for high-dimensional spatiotemporal processes, this approach can be computationally expensive. Che et al. (2018); Shukla and Marlin (2018) modifies the hidden states with exponential decay. GRU-ODE-Bayes proposed by De Brouwer et al. (2019) introduces a continuous-time version of GRU and a Bayesian update network capable of handling sporadic observations. However, Mozer et al.

(2017) shows that there is no significant benefit of using continuous-time RNN for discrete event data. Special treatment is still needed for modeling unevenly sampled events.

**Deep Point Process.**    Point process is well-studied in statistics (Moller and Waagepetersen, 2003; Daley and Vere-Jones, 2007; Reinhart et al., 2018). Deep point process couples deep learning with point process and has received considerable attention. For example, neural Hawkes process applies RNNs to approximate the temporal intensity function (Du et al., 2016; Mei and Eisner, 2017; Xiao et al., 2017; Zhang et al., 2020), and (Zuo et al., 2020) employs Transformers. (Shang and Sun, 2019) integrates graph convolution structure. However, all existing works focus on temporal point processes without spatial modeling. For datasets with spatial information, they discretize the space and treat them as discrete "markers". Okawa et al. (2019) extends Du et al. (2016) for spatiotemporal event prediction but they only predict the density instead of the next location and time of the event. Zhu et al. (2019) parameterizes the spatial kernel with a neural network embedding without consider the temporal sequence. Recently, Chen et al. (2021) propose neural spatiotemporal point process (NSTPP) which combines continuous-time neural networks with continuous-time normalizing flows to parameterize spatiotemporal point processes. However, this approach is quite computationally expensive, which requires evaluating the ODE solver for multiple time steps.

## 4. Experiments

We evaluate `DeepSTPP` for spatiotemporal prediction using both synthetic and real-world data.

**Baselines**    We compare `DeepSTPP` with the state-of-the-art models, including

- Spatiotemporal Hawkes Process (MLE) (Reinhart et al., 2018): it learns a spatiotemporal parametric intensity function using maximum likelihood estimation, see derivation in Appendix A.3.

- Recurrent Marked Temporal Point Process (RMTPP) (Du et al., 2016): it uses GRU to model the temporal intensity function. We modify this model to take spatial location as marks.

- Neural Spatiotemporal Point Process (NSTPP) Chen et al. (2021): a neural point process model that parameterizes the spatial PDF and temporal intensity with continuous-time normalizing flows. Specifically, we use Jump CNF as it is a better fit for Hawkes processes.

All models are implemented in PyTorch, trained using the Adam optimizer. We set the number of representative points to be 100. The details of the implementation are deferred to the Appendix C.1. For the baselines, we use the authors' original repositories whenever possible.

**Datasets.**    We simulated two types of STPPs: spatiotemporal Hawkes process (STH) and spatiotemporal self-correcting process (STSC) . For both STPPs, we generate three synthetic datasets, each with a different parameter setting, denoted as DS1, DS2, and DS3 in the tables. We also derive and implement efficient algorithms for simulating STPPs based on Ogata's thinning algorithm Ogata (1981). We view the simulator construction as an independent contribution from this work. The details of the simulation can be found in Appendix B. We use two real-world spatiotemporal event datasets from NSTPP Chen et al. (2021) to benchmark the performance.

- **Earthquakes Japan**: catalog earthquakes data including the location and time of all earthquakes in Japan from 1990 to 2020 with magnitude of at least 2.5 from the U.S. Geological

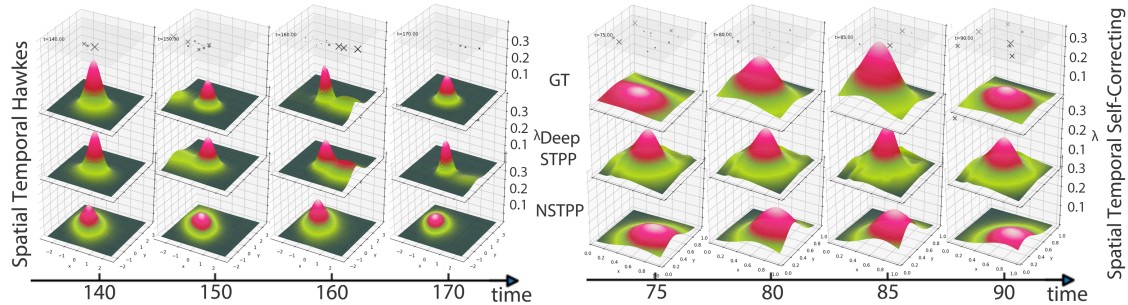

Figure 3: Ground-truth and learned intensity on two synthetic data. **Top**: ground-truth; **Middle**: learned intensity by our `DeepSTPP` model. **Bottom**: learned conditional intensity by NSTPP. 'X's refer to event history, where smaller 'X' refers to larger time difference.

Survey. There are in total 1,050 sequences. The number of events per sequences ranges between 19 to 545 [1].

- **COVID-19**: daily county level COVID-19 cases data in New Jersey state published by The New York Times. There are 1,650 sequences and the number of events per sequences ranges between 7 to 305.

For both synthetic data and real-world data, we partition long event sequences into non-overlapping subsequences according to a fixed time range $T$. The targets are the last event, and the input is the rest of the events. The number of input events varies across subsequences. For each dataset, we split each into train/val/test sets with the ratio of 8:1:1. All results are the average of 3 runs.

Table 1: Test log likelihood (LL) and Hellinger distance of distribution (HD) on synthetic data (LL higher is better, HD lower is better). Comparison between ours and NSTPP on synthetic datasets from two type of spatiotemporal point processes.

| | Spatiotemporal Hawkes process | | | | | | Spatiotemporal Self Correcting process | | | | | |
|---|---|---|---|---|---|---|---|---|---|---|---|---|
| | DS1 | | DS2 | | DS3 | | DS1 | | DS2 | | DS3 | |
| | LL | HD | LL | HD | LL | HD | LL | HD | LL | HD | LL | HD |
| DeepSTPP (ours) | **-3.8420** | **0.0033** | **-3.1142** | **0.4920** | **-3.6327** | **0.0908** | **-1.2248** | **0.2348** | **-1.4915** | **0.1813** | **-1.3927** | **0.2075** |
| NSTPP | -5.3110 | 0.5341 | -4.8564 | 0.5849 | -3.7366 | 0.1498 | -2.0759 | 0.5426 | -2.3612 | 0.3933 | -3.0599 | 0.3097 |

### 4.1. Synthetic Experiment Results

For synthetic data, we know the ground truth intensity function. We compare our method with the best possible estimator: maximum likelihood estimator (MLE), as well as the NSTPP model. The MLE is learned by optimizing the log-likelihood using the BFGS algorithm. RMTPP can only learn the temporal intensity thus is not included in this comparison.

**Predictive log-likelihood.** Table 1 shows the comparison of the predictive distribution for space and time. We report Log Likelihood (LL) of $f(\mathbf{s}, t|\mathcal{H}_t)$ and the Hellinger Distance (HD) between the predictive distributions and the ground truth averaged over time.

---

1. The statistics differ slightly from the original paper due to updates in the data source.

On both the STH and STSC datasets with different parameter settings, `DeepSTPP` outperform the baseline NSTPP in terms of LL and HD. It shows that `DeepSTPP` can estimate the spatiotemporal intensity more accurately for point processes with unknown parameters.

Table 2: Estimated $\lambda^*(t)$ MAPE on synthetic data

|  | STH | | | STSC | | |
|---|---|---|---|---|---|---|
|  | DS1 | DS2 | DS3 | DS1 | DS2 | DS3 |
| `DeepSTPP` | 3.33 | 369.44 | 11.30 | **7.84** | **3.22** | 20.98 |
| NSTPP | 53.41 | 17.69 | 3.85 | 99.99 | 39.33 | 37.39 |
| RMTPP | 263.83 | 729.78 | **0.62** | 45.55 | 21.26 | 37.46 |
| MLE | **2.98** | **11.30** | 4.38 | 27.38 | 18.20 | **20.01** |

**Temporal intensity estimate.** Table 2 shows the mean absolute percentage error (MAPE) between the models' estimated temporal intensity and the ground truth $\lambda^\star(t)$ over a short sampled range. On the STH datasets, since MLE has the correct parametric form, it is the theoretical optimum. Compared to baselines, `DeepSTPP` generally obtained the same or lower MAPE. It shows that joint spatiotemporal modeling also improve the performance of temporal prediction.

**Intensity visualization.** Figure 3 visualizes the learned space-time intensity and the ground truth for STH and STSC, providing strong evidence that `DeepSTPP` can correctly learn the underlying dynamics of the spatiotemporal events. Especially, NSTPP has difficulty in modeling the complex dynamics of the multimodal distribution such as the spatiotemporal Hawkes process. NSTPP sometimes produces overly smooth intensity surfaces, and lost most of the details at the peak. In contrast, our `DeepSTPP` can better fit the multimodal distribution through the form of kernel summation and obtain more accurate intensity functions.

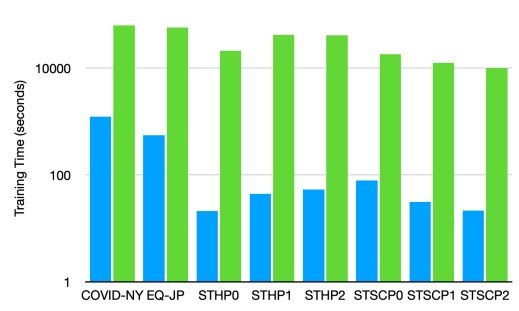

Figure 4: Log train time comparison on all datasets

**Computational efficiency.** Figure 4 provides the run time comparison for the training between `DeepSTPP` and NSTPP for 100 epochs. To ensure a fair comparison, all experiments are conducted on 1 GTX 1080 Ti with Intel Core i7-4770 and 64 GB RAM. Our method is 100 times faster than NSTPP in training. It is mainly because our spatiotemporal kernel formulation has a close form of integration, which bypasses the complex and cumbersome numerical integration.

### 4.2. Real-World Experiment Results

For real-world data evaluation, we report the conditional spatial and temporal log-likelihoods, i.e., $\log f^*(\mathbf{s}|t)$ and $\log f^*(t)$, of the final event given the input events, respectively. The total log-likelihood, $\log f^*(s,t)$, is the summation of the two values.

Table 3: Test log likelihood (LL) comparison for space and time on real-world data over 3 runs.

| LL | COVID-19 NY | | Earthquake JP | |
|---|---|---|---|---|
|  | Space | Time | Space | Time |
| `DeepSTPP` | $-0.1150_{\pm 0.0109}$ | $2.4583_{\pm 0.0008}$ | $\mathbf{-4.4025}_{\pm 0.0128}$ | $\mathbf{0.4173}_{\pm 0.0014}$ |
| NSTPP | $\mathbf{-0.0798}_{\pm 0.0433}$ | $\mathbf{2.6364}_{\pm 0.0111}$ | $-4.8141_{\pm 0.1165}$ | $0.3192_{\pm 0.0124}$ |
| RMTPP | - | $2.4476_{\pm 0.0039}$ | - | $0.3716_{\pm 0.0077}$ |

**Predictive performances.** As our model is probabilistic, we compare

against baselines models on the test predictive LL for space and time separately in Table 3. RMTPP can only produce temporal intensity thus we only include the time likelihood. We observe that `DeepSTPP` outperforms NSTPP most of the time in terms of accuracy. It takes only half of the time to train, as shown in Figure 4. Furthermore, we see that STPP models (first three rows) achieve higher LL compared with only modeling the time (RMTPP). It suggests the additional benefit of joint spatiotemporal modeling to increases the time prediction ability.

**Ablation study** We conduct ablation studies on the model design. Our model assumes a global latent process $z$ that governs the parameters $\{w_i, \beta_i, \gamma_i\}$ with separate decoders. We examine other alternative designs experimentally. (1) *Shared decoders*: We use one shared decoder to gener-

Table 4: Test LL for alternative model designs over 3 runs

| (higher the better) | COVID-19 NY | | STH DS2 | |
|---|---|---|---|---|
| | Space | Time | Space | Time |
| Shared decoders | $-0.1152_{\pm 0.0142}$ | $2.4581_{\pm 0.0030}$ | $-2.4397_{\pm 0.0170}$ | $\mathbf{-0.6060}_{\pm 0.0381}$ |
| Separate processes | $\mathbf{-0.1057}_{\pm 0.0140}$ | $2.4561_{\pm 0.0048}$ | $-2.4291_{\pm 0.0123}$ | $-0.7022_{\pm 0.0050}$ |
| LSTM encoder | $-0.1162_{\pm 0.0102}$ | $2.4554_{\pm 0.0035}$ | $-2.4331_{\pm 0.0174}$ | $-0.6845_{\pm 0.0252}$ |
| `DeepSTPP` | $-0.1150_{\pm 0.0109}$ | $\mathbf{2.4583}_{\pm 0.0008}$ | $\mathbf{-2.4289}_{\pm 0.0102}$ | $-0.6853_{\pm 0.0145}$ |

ate model parameters. Shared decoders input the sampled $z$ to one decoder and partition its output to generate model parameters.(2) *Separate process*: We assume that each of the $\{w_i, \beta_i, \gamma_i\}$ follows a separate latent process and we sample them separately. Separate processes use three sets of means and variances to sample $\{w_i, \beta_i, \gamma_i\}$ separately. (3) *LSTM encoder*: We replace the Transformer encoder with a LSTM module.

As shown in Table 4, we see that (1) *Shared decoders* decreases the number of parameters but reduces the performance. (2) *Separate process* largely increases the number of parameters but has negligible influences in test log-likelihood. (3) *LSTM encoder*: changing the encoder from Transformer to LSTM also results in slightly worse performance. Therefore, we validate the design of DeepNSTPP: we assume all distribution parameters are governed by one single hidden stochastic process with separate decoders and a Transformer as encoder.

## 5. Conclusion

We propose a family of deep dynamics models for irregularly sampled spatiotemporal events. Our model, Deep Spatiotemporal Point Process (`DeepSTPP`), integrates a principled spatiotemporal point process with deep neural networks. We derive a tractable inference procedure by modeling the space-time intensity function as a composition of kernel functions and a latent stochastic process. We infer the latent process with neural networks following the variational inference procedure. Using synthetic data from the spatiotemporal Hawkes process and self-correcting process, we show that our model can learn the spatiotemporal intensity accurately and efficiently. We demonstrate superior forecasting performance on many real-world benchmark spatiotemporal event datasets. Future work include further considering the mutual-exciting structure in the intensity function, as well as modeling multiple heterogeneous spatiotemporal processes simultaneously.

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

## Appendix A. Model Details

### A.1. Spatiotemporal Point Process Derivation

**Conditional Density.**    The intensity function and probability density function of STPP is related:

$$
\begin{aligned}
f(\mathbf{s}, t | \mathcal{H}_t) &= \frac{\lambda^*(\mathbf{s}, t)}{1 - F^*(\mathbf{s}, t)} \\
&= \lambda^*(\mathbf{s}, t) \exp\left(-\int_{\mathcal{S}} \int_{t_n}^{t} \lambda^*(\mathbf{s}, \tau) d\tau d\mathbf{s}\right) \\
&= \lambda^*(\mathbf{s}, t) \exp\left(-\int_{t_n}^{t} \lambda^*(\tau) d\tau\right)
\end{aligned}
$$

The last equation uses the relation that $\lambda^*(\mathbf{s}, t) = \lambda^*(t) f(\mathbf{s}|t)$, according Daley and Vere-Jones (2007) Chapter 2.3 (4). Here $\lambda^*(t)$ is the time intensity and $f^*(\mathbf{s}|t) := f(\mathbf{s}|t, \mathcal{H}_t)$ is the spatial PDF that the next event will be at location $\mathbf{s}$ given time $t$. According to Daley and Vere-Jones (2007) Chapter 15.4, we can also view STPP as a type of TPP with continuous (spatial) marks,

**Likelihood.**    Given a STPP, the log-likelihood of observing a sequence $\mathcal{H}_t = \{(\mathbf{s}_1, t_1), (\mathbf{s}_2, t_2), ...(\mathbf{s}_n, t_n)\}_{t_n \leq t}$ is given by:

$$
\begin{aligned}
\mathcal{L}(\mathcal{H}_{t_n}) &= \log\left[\prod_{i=1}^{n} f(\mathbf{s}_i, t_i | \mathcal{H}_{t_{i-1}})(1 - F^*(\mathbf{s}, t))\right] \\
&= \sum_{i=1}^{n} \left[\log \lambda^*(\mathbf{s}_i, t_i) - \int_{\mathcal{S}} \int_{t_{i-1}}^{t_i} \lambda^*(\tau) d\tau d\mathbf{s}\right] + \log(1 - F^*(\mathbf{s}, t)) \\
&= \sum_{i=1}^{n} \log \lambda^*(\mathbf{s}_i, t_i) - \int_{\mathcal{S}} \int_{0}^{t_n} \lambda^*(\mathbf{s}, \tau) d\tau - \int_{\mathcal{S}} \int_{t_n}^{T} \lambda^*(\mathbf{s}, \tau) d\tau \\
&= \sum_{i=1}^{n} \log \lambda^*(\mathbf{s}_i, t_i) - \int_{\mathcal{S}} \int_{0}^{T} \lambda^*(\mathbf{s}, \tau) d\tau \\
&= \sum_{i=1}^{n} \log \lambda^*(t_i) + \sum_{i=1}^{n} \log f^*(\mathbf{s}_i | t_i) - \int_{0}^{T} \lambda^*(\tau) d\tau
\end{aligned}
$$

**Inference.**    With a trained STPP and a sequence of history events, we can predict the next event timing and location using their expectations, which evaluate to

$$
\begin{aligned}
\mathbb{E}[t_{n+1} | \mathcal{H}_{t_n}] &= \int_{t_n}^{\infty} t \int_{\mathcal{S}} f(\mathbf{s}, t | \mathcal{H}_{t_n}) d\mathbf{s} dt = \int_{t_n}^{\infty} t \exp\left(-\int_{t_n}^{t} \lambda^*(\tau) d\tau\right) \int_{\mathcal{S}} \lambda^*(\mathbf{s}, t) d\mathbf{s} dt, \\
&= \int_{t_n}^{\infty} t \exp\left(-\int_{t_n}^{t} \lambda^*(\tau) d\tau\right) \lambda^*(t) dt
\end{aligned}
\tag{12}
$$

The predicted location for the next event is:

$$
\begin{aligned}
\mathbb{E}[\mathbf{s}_{n+1} | \mathcal{H}_{t_n}] &= \int_{t_n}^{\infty} \mathbf{s} \int_{\mathcal{S}} \lambda^*(\mathbf{s}, t) \exp\left(-\int_{t_n}^{t} \lambda^*(\mathbf{s}, \tau) d\tau\right) d\mathbf{s} dt \\
&= \int_{t_n}^{\infty} \exp\left(-\int_{t_n}^{t} \lambda^*(\tau) d\tau\right) \int_{\mathcal{S}} \mathbf{s} \lambda^*(\mathbf{s}, t) d\mathbf{s} dt
\end{aligned}
\tag{13}
$$

**Computational Complexity.** It is worth noting that both learning and inference require conditional intensity. If the conditional intensity has no analytic formula, then we need to compute numerical integration over $\mathcal{S}$. Then, evaluating the likelihood or either expectation requires at least triple integral. Note that $\mathbb{E}[t_i|\mathcal{H}_{t_{i-1}}]$ and $\mathbb{E}[\mathbf{s}_i|\mathcal{H}_{t_{i-1}}]$ actually are sextuple integrals, but we can memorize all $\lambda^*(\mathbf{s}, t)$ from $t = t_{i-1}$ to $t \gg t_{i-1}$ to avoid re-compute the intensities. However, memorization leads to high space complexity. As a result, we generally want to avoid an intractable conditional intensity in the model.

### A.2. Deep Spatiotemporal Point process (`DeepSTPP`) Derivation

**PDF Derivation** The model design of `DeepSTPP` enjoys a closed form formula for the PDF. First recall that

$$f^*(t) = \lambda^*(t) \exp\left(-\int_{t_n}^t \lambda^*(\tau)d\tau\right)$$

Also notice that $f^*(\mathbf{s}, t) = f^*(\mathbf{s}|t)f^*(t)$, $\lambda^*(\mathbf{s}, t) = f^*(\mathbf{s}|t)\lambda^*(t)$ and $\lambda^*(t) = \dfrac{f^*(t)}{1 - F^*(t)}$ .

Therefore

$$f^*(\mathbf{s}, t) = f^*(\mathbf{s} \mid t)f^*(t)$$

$$= f^*(\mathbf{s} \mid t)\lambda^*(t) \exp\left(-\int_{t_n}^t \lambda^*(\tau)d\tau\right)$$

$$= \lambda^*(\mathbf{s}, t) \exp\left(-\int_{t_n}^t \lambda^*(\tau)d\tau\right)$$

For `DeepSTPP`, the spatiotemporal intensity is

$$\lambda^*(\mathbf{s}, t) = \sum_i w_i \exp(-\beta_i(t - t_i))k_s(\mathbf{s} - \mathbf{s}_i)$$

The temporal intensity simply removes the $k_s$ (which integrates to one). The bandwidth doesn't matter.

$$\lambda^*(t) = \sum_i w_i \exp(-\beta_i(t - t_i))$$

Integrate $\lambda^*(\tau)$ yields

$$\int \lambda^*(\tau)d\tau = -\sum_i \frac{w_i}{\beta_i} \exp(-\beta_i(\tau - t_i)) + C$$

Note that deriving the $\exp$ would multiply the coefficient $-\beta_i$.

The definite integral is

$$\int_{t_n}^t \lambda^*(\tau)d\tau = -\sum_i \frac{w_i}{\beta_i}[\exp(-\beta_i(t - t_i)) - \exp(-\beta_i(t_n - t_i))]$$

Then replacing the integral in the original formula yields

$$f^*(\mathbf{s}, t) = \lambda^*(\mathbf{s}, t) \exp\left(-\int_{t_n}^{t} \lambda^*(\tau)d\tau\right)$$

$$= \lambda^*(\mathbf{s}, t) \exp\left(\sum_i \frac{w_i}{\beta_i}[\exp(-\beta_i(t - t_i)) - \exp(-\beta_i(t_n - t_i))]\right)$$

The temporal kernel function $k_t(t, t_i) = \exp(-\beta_i(t - t_i))$, we reach the closed form formula.

**Inference**   The expectation of the next event time is

$$\mathbb{E}^*[t_i] = \int_{t_{i-1}}^{\infty} t f^*(t)dt = \int_{t_n}^{\infty} t \lambda^*(t) \exp\left(-\int_{t_{i-1}}^{t} \lambda^*(\tau)d\tau\right) dt$$

where the inner integral has a closed form. It requires 1D numerical integration.

Given the predicted time $\bar{t}_i$, the expectation of the space can be efficiently approximated by

$$\mathbb{E}^*[\mathbf{s}_i] \approx \mathbb{E}^*[\mathbf{s}_i|\bar{t}_i] = \sum_{i'<i} \alpha^{-1} w_{i'} k_t(\bar{t}_i, t_{i'})\mathbf{s}_{i'}$$

where $\alpha = \sum_{i'<i} w_{i'} k_t(\bar{t}_i, t_{i'})$ is a normalize coefficient.

### A.3. Spatiotemporal Hawkes Process Derivation

**Spatiotemporal Hawkes process (STHP).**   Spatiotemporal Hawkes (or self-exciting) process is one of the most well-known STPPs. It assumes every past event has an additive, positive, decaying, and spatially local influence over future events. Such a pattern resembles neuronal firing and earthquakes.

Spatiotemporal Hawkes is characterized by the following intensity function (Reinhart et al., 2018):

$$\lambda^*(\mathbf{s}, t) := \mu g_0(\mathbf{s}) + \sum_{i:t_i<t} g_1(t, t_i)g_2(\mathbf{s}, \mathbf{s}_i) : \mu > 0 \tag{14}$$

where $g_0(\mathbf{s})$ is the probability density of a distribution over $\mathcal{S}$, $g_1$ is the triggering kernel and is often implemented as the exponential decay function, $g_1(\Delta t) := \alpha \exp(-\beta \Delta t) : \alpha, \beta > 0$, and $g_2(\mathbf{s}, \mathbf{s}_i)$ is the density of an unimodal distribution over $\mathcal{S}$ centered at $\mathbf{s}_i$.

**Maximum Likelihood.**   For spatiotemporal Hawkes process, we pre-specified the model kernels $g_0(\mathbf{s})$ and $g_2(\mathbf{s}, \mathbf{s}_j)$ to be Gaussian:

$$g_0(\mathbf{s}) := \frac{1}{2\pi}|\Sigma_{g0}|^{-\frac{1}{2}} \exp\left(-\frac{1}{2}(\mathbf{s} - \mathbf{s}_\mu)\Sigma_{g0}^{-1}(\mathbf{s} - \mathbf{s}_\mu)^T\right) \tag{15}$$

$$g_2(\mathbf{s}, \mathbf{s}_j) := \frac{1}{2\pi}|\Sigma_{g2}|^{-\frac{1}{2}} \exp\left(-\frac{1}{2}(\mathbf{s} - \mathbf{s}_j)\Sigma_{g2}^{-1}(\mathbf{s} - \mathbf{s}_j)^T\right) \tag{16}$$

Specifically for the STHP, the second term in the STPP likelihood evaluates to

$$\int_0^T \lambda^*(\tau)d\tau = \mu T + \alpha \int_0^T \int_0^\tau e^{-\beta(\tau-u)} dN(u)d\tau$$

$$(0 \le u \le \tau, 0 \le \tau \le T) \to (u \le \tau \le T, 0 \le u \le T)$$

$$= \mu T + \alpha \int_0^T \int_u^T e^{-\beta(\tau-u)} d\tau dN(u)$$

$$= \mu T - \frac{\alpha}{\beta} \int_0^T \left[ e^{-\beta(T-u)} - 1 \right] dN(u)$$

$$= \mu T - \frac{\alpha}{\beta} \sum_{i=0}^N \left[ e^{-\beta(T-t_i)} - 1 \right]$$

Finally, the STHP log-likelihood is

$$\mathcal{L} = \sum_{i=1}^n \log \lambda^*(\mathbf{s}_i, t_i) - \mu T + \frac{\alpha}{\beta} \sum_{i=0}^N \left[ e^{-\beta(T-t_i)} - 1 \right]$$

This model has 11 scalar parameters: 2 for $\mathbf{s}_\mu$, 3 for $\Sigma_{g0}$, 3 for $\Sigma_{g2}$, $\alpha, \beta$, and $\mu$. We directly estimate $\mathbf{s}_\mu$ as the mean of $\{s_i\}_0^n$, and then estimate the other 9 parameters by minimizing the negative log-likelihood using the BFGS algorithm. $T$ in the likelihood function is treated as $t_n$.

**Inference**    Based on the general formulas in Appendix A.1, and also note that for an STHP,

$$\int_{t_{i-1}}^t \lambda^*(\tau)d\tau = \int_0^t \lambda^*(\tau)d\tau - \int_0^{t_{i-1}} \lambda^*(\tau)d\tau$$

$$= \left\{ \mu t - \frac{\alpha}{\beta} \sum_{j=0}^{i-1} \left[ e^{-\beta(t-t_j)} - 1 \right] \right\} - \left\{ \mu t_{i-1} - \frac{\alpha}{\beta} \sum_{j=0}^{i-1} \left[ e^{-\beta(t_{i-1}-t_j)} - 1 \right] \right\}$$

$$= \mu(t - t_{i-1}) - \frac{\alpha}{\beta} \sum_{j=0}^{i-1} \left[ e^{-\beta(t-t_{i-1}+t_{i-1}-t_j)} - e^{-\beta(t_{i-1}-t_j)} \right]$$

$$= \mu(t - t_{i-1}) - \frac{\alpha}{\beta} \left( e^{-\beta(t-t_{i-1})} - 1 \right) \sum_{j=0}^{i-1} \left[ e^{-\beta(t_{i-1}-t_j)} \right] \text{ and}$$

$$\int_{\mathcal{S}} \mathbf{s}\mu g_2(\mathbf{s}, \mathbf{s}_\mu)d\mathbf{s} = \mu \mathbf{s}_\mu$$

$$\int_{\mathcal{S}} \mathbf{s} \sum_{i=0}^n g_1(t, t_i)g_2(\mathbf{s}, \mathbf{s}_i)d\mathbf{s} = \sum_{i=0}^n g_1(t, t_i) \int_{\mathcal{S}} \mathbf{s}g_2(\mathbf{s}, \mathbf{s}_i)d\mathbf{s} = \sum_{i=0}^n g_1(t, t_i)\mathbf{s}_i$$

$$\int_{\mathcal{S}} \mathbf{s}\lambda^*(\mathbf{s}, t)d\mathbf{s} = \mu \mathbf{s}_\mu + \sum_{i=0}^n g_1(t, t_i)\mathbf{s}_i,$$

we have

$$\mathbb{E}[t_i|\mathcal{H}_{t_{i-1}}] = \int_{t_{i-1}}^{\infty} t \left( \mu + \alpha \sum_{j=0}^{i-1} e^{-\beta(t-t_j)} \right)$$
$$\exp\left( \frac{\alpha}{\beta} \left( e^{-\beta(t-t_{i-1})} - 1 \right) \sum_{j=0}^{i-1} \left[ e^{-\beta(t_{i-1}-t_j)} \right] - \mu(t - t_{i-1}) \right) dt \text{ and}$$

$$\mathbb{E}[\mathbf{s}_i|\mathcal{H}_{t_{i-1}}] = \int_{t_{i-1}}^{\infty} \left( \mu \mathbf{s}_\mu + \alpha \sum_{j=0}^{i-1} e^{-\beta(t-t_j)} \mathbf{s}_j \right)$$
$$\exp\left( \frac{\alpha}{\beta} \left( e^{-\beta(t-t_{i-1})} - 1 \right) \sum_{j=0}^{i-1} \left[ e^{-\beta(t_{i-1}-t_j)} \right] - \mu(t - t_{i-1}) \right) dt$$

Both require only 1D numerical integration.

**Spatiotemporal Self-Correcting process (STSCP).**   A lesser-known example is self-correcting spatiotemporal point process Isham and Westcott (1979). It assumes that the background intensity increases with a varying speed at different locations, and the arrival of each event reduces the intensity nearby. The next event is likely to be in a high-intensity region with no recent events.

Spatiotemporal self-correcting process is capable of modeling some regular event sequences, such as an alternating home-to-work travel sequence. It has the following intensity function:

$$\lambda^*(\mathbf{s}, t) = \mu \exp\left( g_0(\mathbf{s})\beta t - \sum_{i:t_i < t} \alpha g_2(\mathbf{s}, \mathbf{s}_i) \right) : \alpha, \beta, \mu > 0 \tag{17}$$

Here $g_0(\mathbf{s})$ is the density of a distribution over $\mathcal{S}$, and $g_2(\mathbf{s}, \mathbf{s}_i)$ is the density of an unimodal distribution over $\mathcal{S}$ centered at $\mathbf{s}_i$.

## Appendix B.  Simulation Details

In this appendix, we discuss a general algorithm for simulating any STPP, and a specialized algorithm for simulating an STHP. Both are based on an algorithm for simulating any TPP.

### B.1.  TPP Simulation

The most widely used technique to simulate a temporal point process is Ogata's modified thinning algorithm, as shown in Algorithm 1 Daley and Vere-Jones (2007) It is a rejection technique; it samples points from a stationary Poisson process whose intensity is always higher than the ground truth intensity, and then randomly discards some samples to get back to the ground truth intensity.

The algorithm requires picking the forms of $M^*(t)$ and $L^*(t)$ such that

$$\sup(\lambda^*(t + \Delta t), \Delta t \in [0, L(t)]) \leq M^*(t).$$

In other words, $M^*(t)$ is an upper bound of the actual intensity in $[t, t + L(t)]$. It is noteworthy that if $M^*(t)$ is chosen to be too high, most sampled points would be rejected and would lead to an inefficient simulation.

When simulating a process with decreasing inter-event intensity, such as the Hawkes process, $M^*(t)$ and $L^*(t)$ can be simply chosen to be $\lambda^*(t)$ and $\infty$. When simulating a process with increasing inter-event intensity, such as the self-correcting process, $L^*(t)$ is often empirically chosen to be $2/\lambda^*(t)$, since the next event is very likely to arrive before twice the mean interval length at the beginning of the interval. $M^*(t)$ is therefore $\lambda^*(t + L^*(t))$.

---

**Algorithm 1** Ogata Modified Thinning Algorithm for Simulating a TPP

---

1: **Input:** Interval $[0, T]$, model parameters
2: $t \leftarrow 0, \mathcal{H} \leftarrow \emptyset$
3: **while true do**
4:     Compute $m \leftarrow M(t|\mathcal{H})$ , $l \leftarrow L(t|\mathcal{H})$
5:     Draw $\Delta t \sim \mathrm{Exp}(m)$ (exponential distribution with mean $1/m$)
6:     **if** $t + \Delta t > T$ **then**
7:         **return** $\mathcal{H}$
8:     **end if**
9:     **if** $\Delta t > l$ **then**
10:         $t \leftarrow t + l$
11:     **else**
12:         $t \leftarrow t + \Delta t$
13:         Compute $\lambda = \lambda^*(t)$
14:         Draw $u \sim \mathrm{Unif}(0, 1)$
15:         **if** $\lambda/m > u$ **then**
16:             $\mathcal{H} = \mathcal{H} \cup t$
17:         **end if**
18:     **end if**
19: **end while**=0

---

## B.2. STPP Simulation

It has been mentioned in Section 2.1 that an STPP can be seen as attaching the locations sampled from $f^*(\mathbf{s}|t)$ to the events generated by a TPP. Simulating an STPP is basically adding one step to Algorithm 1: sample a new location from $f^*(\mathbf{s}|t)$ after retaining a new event at $t$.

As for a spatiotemporal self-correcting process, neither $f^*(\mathbf{s}, t)$ nor $\lambda^*(t)$ has a closed form, so the process's spatial domain has to be discretized for simulation. $\lambda^*(t)$ can be approximated by $\sum_{\mathbf{s} \in \mathcal{S}} \lambda^*(\mathbf{s}, t)/|\mathcal{S}|$, where $\mathcal{S}$ is the set of discretized coordinates. $L^*(t)$ and $M^*(t)$ are chosen to be $2/\lambda^*(t)$ and $\lambda^*(t + L^*(t))$. Since $f^*(\mathbf{s}|t)$ is proportional to $\lambda^*(\mathbf{s}, t)$, sampling a location from $f^*(\mathbf{s}|t)$ is implemented as sampling from a multinomial distribution whose probability mass function is the normalized $\lambda^*(\mathbf{s}, t)$.

## B.3. STHP Simulation

To simulate a spatiotemporal Hawkes process with Gaussian kernel, we mainly followed an efficient procedure proposed by Zhuang (2004), that makes use of the clustering structure of the Hawkes process and thus does not require repeated calculations of $\lambda^*(\mathbf{s}, t)$.

---

**Algorithm 2** Simulating spatiotemporal Hawkes process with Gaussian kernel

---

1: Generate the background events $G^{(0)}$ with the intensity $\lambda^*(\mathbf{s}, t) = \mu g_0(\mathbf{s})$, i.e., simulate a homogenous Poisson process $\mathrm{Pois}(\mu)$ and sample each event's location from a bivariate Gaussian distribution $\mathcal{N}(\mathbf{s}_\mu, \Sigma)$
2: $\ell = 0, S = G^{(\ell)}$
3: **while** $G^\ell \neq \emptyset$ **do**
4:    **for** $i \in G^\ell$ **do**
5:       Simulate event $i$'s offsprings $O_i^{(\ell)}$ with the intensity $\lambda^*(\mathbf{s}, t) = g_1(t, t_i) g_2(\mathbf{s}, \mathbf{s}_i)$, i.e., simulate a non-homogenous stationary Poisson process $\mathrm{Pois}(g_1(t, t_i))$ by **Algorithm 1** and sample each event's location from a bivariate Gaussian distribution $\mathcal{N}(\mathbf{s}_i, \Sigma)$
6:       $G^{(\ell+1)} = \bigcup_i O_i^{(\ell)}, S = S \cup G^{(\ell+1)}, \ell = \ell + 1$
7:    **end for**
8: **end while**
9: **return** $S$ =0

---

Table 5: Parameter settings for the synthetic dataset

|  |  | $\alpha$ | $\beta$ | $\mu$ | $\Sigma_{g0}$ | $\Sigma_{g2}$ |
|---|---|---|---|---|---|---|
| ST-Hawkes | DS1 | .5 | 1 | .2 | [.2 0; 0 .2] | [0.5 0; 0 0.5] |
|  | DS2 | .5 | .6 | .15 | [5 0; 0 5] | [.1 0; 0 .1] |
|  | DS3 | .3 | 2 | 1 | [1 0; 0 1] | [.1 0; 0 .1] |
| ST-Self Correcting | DS1 | .2 | .2 | 1 | [1 0; 0 1] | [0.85 0; 0 0.85] |
|  | DS2 | .3 | .2 | 1 | [.4 0; 0 .4] | [.3 0; 0 .3] |
|  | DS3 | .4 | .2 | 1 | [.25 0; 0 .25] | [.2 0; 0 .2] |

## B.4. Parameter Settings

For the synthetic dataset, we pre-specified both the STSCP's and the STHP's kernels $g_0(\mathbf{s})$ and $g_2(\mathbf{s}, \mathbf{s}_j)$ to be Gaussian:

$$g_0(\mathbf{s}) := \frac{1}{2\pi} |\Sigma_{g0}|^{-\frac{1}{2}} \exp\left(-\frac{1}{2}(\mathbf{s} - [0, 0])\Sigma_{g0}^{-1}(\mathbf{s} - [0, 0])^T\right)$$

$$g_2(\mathbf{s}, \mathbf{s}_j) := \frac{1}{2\pi} |\Sigma_{g2}|^{-\frac{1}{2}} \exp\left(-\frac{1}{2}(\mathbf{s} - \mathbf{s}_j)\Sigma_{g2}^{-1}(\mathbf{s} - \mathbf{s}_j)^T\right)$$

The STSCP is defined on $\mathcal{S} = [0, 1] \times [0, 1]$, while the STHP is defined on $\mathcal{S} = \mathbb{R}^2$. The STSCP's kernel functions are normalized according to their cumulative probability on $\mathcal{S}$. Table 5 shows the simulation parameters. The STSCP's spatial domain is discretized as an $101 \times 101$ grid during the simulation.

## Appendix C. Experiment Details

In this section, we include experiment configurations and some additional experiment results.

## C.1. Model Setup Details

For a better understanding of `DeepSTPP`, we list out the detailed hyperparameter settings in Table 6. We use the same set of hyperparameters across all datasets.

| Name | Value | Description |
|---|---|---|
| Optimizer | Adam | Optimizer of the Transformer-VAE is set to Adam |
| Learning rate | - | 0.01(Synthetic) / 0.015(Real World) |
| Momentum | 0.9 | - |
| Epoch | 200 | Train the VAE for 200 epochs for 1 step prediction |
| Batch size | 128 | - |
| Encoder: `nlayers` | 3 | Encoder is composed of a stack of 3 identical Transformer layers |
| Encoder: `nheads` | 2 | Number of attention heads in each Transformer layer |
| Encoder: $d_{\text{model}}$ | 128 | 3-tuple history is embedded to 128-dimension before fed into encoder |
| Encoder: $d_{\text{hidden}}$ | 128 | Dimension of the feed-forward network model in each Transformer layer |
| Positional Encoding | Sinusoidal | Default encoding scheme in Vaswani et al. (2017) |
| Decoder: $d_{\text{hidden}}$ | 128 | Decoders for $w_i, \beta_i,$ and $\gamma_i$ are all MLPs with 2 hidden layers whose dim = 128 |
| $d_z$ | 128 | Dimension of the latent variable $z$ as shown in Figure 2 |
| $J$ | 50 | Number of representative points as described in Section 2.2; 100 during |
| $\beta$ | 1e-3 | Scale factor multiplied to the log-likelihood in VAE loss |

Table 6: Hyperparameter settings for training `DeepSTPP` on all datasets.

