# OpenReview forum: "Neural Point Process for Forecasting Spatiotemporal Events"
_ICLR.cc/2021/Conference — Reject_

### Official Review · AnonReviewer1 · 2020-10-20
**review for #980**

**Rating:** 4
**Confidence:** 4

**Review:**

This work studies the DNN-based spatiotemporal point process model. It points out the drawback of most existing DNN-based point process models: incapability to incorporate the spatio information. Although in statistics, the spatiotemporal point process is capable of capturing events in continuous space and time, such methods are computation expensive. The theoretical analysis is provided, and experimental comparisons are conducted on synthetic and real data.

Strengths: The model setup is reasonable, and the paper writing is easy to follow. The work successfully includes the space information into the DNN-based point process model with a KDE method.

But I recommend rejection of the paper for the reasons below.

Weakness: The major concern is the contribution of the work is not significant enough for publishing on ICLR, because most of the work (in fact, almost all the work on the temporal part) is same as the work “Recurrent marked temporal point processes: Embedding event history to vector. KDD, 2016.” as cited by the submission, e.g. the conditional intensity in Eq.(5). The improvement of the model on spatial information is implemented by introducing a KDE method to learn the spatial pdf. The KDE method has a major drawback that the current spatial pdf does not depend on the current time (eq.(3)) which is counter-intuitive, as in most cases it is natural to decide when it happens first and then locate where it happens, just as the definition of \lambda_(s, t) = f_(s|t)*\lambda__(t) in the submission.

Some specific concerns: in abstract, the author stated “time-conditioned spatial density function”. This is not accurate as it is not conditioned on current time as mentioned above.
The example in the end of sec.3.1 is not appropriate. Because the earthquake observation in the example only contains time and magnitude (no location information), of course it has no advantage over RNN for location prediction.
The kernel for KDE is chosen to be Gaussian kernel, so how to choose the hyperparameters, e.g. \Sigma, for the kernel function? I only saw it is fixed to identity matrix in the experiments, so why choose identity? Any validation experiments?
In sec.4.2, why use the ‘max’ regularizer to avoid w_j>>w_k? why not use the L2 regularizer? is the max regularizer better than L2? why set the hyperparameter \gamma to 1e-4 in experiments? Any validation experiments?
In experiments, why fix w^t and b to 0.1, respectively? If my understanding is right, these two parameters should be learned in the training process.
In experiments, why are only 1 or 2 baseline models considered in Table 1 and 2? What is the performance of other baselines?
In experiments, what is the definition of joint RMSE?


Typo: in “Relationship to TPP”, f^*(s|t)=f(s|t,H_t)
In Eq.(5), h_{i-1}-->h_{i}    H_{t_{i-1}}-->H_{t_{i}}
In eq.(7), E[t_i|H_t_{i-1}]-->E[t_{i+1}|H_t_i]     E[s_i|H_t_{i-1}]-->E[s_{i+1}| H_t_i]
Sec.4.2, S={s_i-j,t_i-j}_{j=1}^L-->S={s_i,t_i}_{i=1}^N  the RNN input is J but the whole observation is N.
In “Baselines”, ofspatiotem --> of spatiotemporal

---

> ### Author Response · Authors · 2020-11-24
> **Response to Reviewer 1: Significance of our contribution and new experiments**
>
> Thank you for your review.
>
> __Contribution of the work is not significant enough for publishing on ICLR...__
>
> A novel temporal component is not the focus of this paper; we clarified this and made explicit reference to Du et al in the revised paper. Our paper’s main goal is to fill the gap of spatial modeling and establish a neural point process framework for spatiotemporal forecasting. The temporal component can be any marked TPP model’s intensity function. Our paper’s another novelty is the accurate simulation of an arbitrary spatiotemporal point process (see Appendix B.2), which is highly non-trivial [Reinhart et al, 2018]. As far as we know, among all spatiotemporal point processes, only the spatiotemporal Hawkes process has well-developed simulation methods [Ogata 1998, Zhuang 2004]. We described a general way of simulating an STPP given its spatiotemporal intensity function, even when its temporal intensity function has no closed-form. In addition, we designed STSCP featuring self-inhibition and demonstrated the method’s validity on simulating an STSCP.  This will serve as a validation tool for the community to evaluate the correctness of various neural models.
>
>
> __The KDE method has a major drawback that the current spatial pdf does not depend on the current time. This is counter-intuitive.__
>
> Thank you for pointing this out. In fact, the KDE method is compatible with any conditioned spatial pdf. The weight of the kernels can have the current time as a part of the input.   We have conducted additional experiments to include the current time into the spatial PDF as described in Sections 4.3.  We added the results for this  and 5.3. However, we found the performance to be worse. The error in the next event timing prediction would accumulate to the error in spatial prediction. While adding current time to spatial PDF makes the model more expressive, the prediction accuracy deteriorates.
>
> __The author stated “time-conditioned spatial density function”. This is not accurate as it is not conditioned on current time as mentioned above.__
>
> Our spatial PDF was designed to be conditioned on the past time. We have also experimented with ``conditioning on current time’’ but did not see performance improvement (see Section 4.3 and 5.3 for details).
>
> __Why fix the covariance to be identity? Why fix $w^t$ and $w^b$ to be 0.1? Why are only 1 or 2 baseline models considered in Table 1 and 2?__
>
> This is a typo and we have fixed it; those variables are not hyperparameters but trainable parameters. Covariance is initialized to be identity and $w^t$ and $w^b$ are initialized to 0.1.
>
> __The example in the end of sec.3.1 is not appropriate.__
>
> We have rephrased the example in Sec 3.1.
>
> __The kernel for KDE is chosen to be Gaussian kernel, so how to choose the hyperparameters?__
>
> The covariance matrix of the Gaussian kernel are also not hyperparameters but trainable parameters.
>
>  __Why are only 1 or 2 baseline models considered in Table 1 and 2? What is the performance of other baselines?__
>
> There are very few models that can learn the spatiotemporal intensity. Most neural TPP models only learn the temporal intensity. To the best of our knowledge, the spatiotemporal Hawkes process MLE learner is the only candidate.

---

### Official Review · AnonReviewer3 · 2020-10-28
**a reasonable model but lack of novelty or careful enough justification: reject**

**Rating:** 4
**Confidence:** 5

**Review:**

The paper proposed a neural point process that learns to predict time and location of an event. The temporal component uses the parametrization of Du et al 2016 and the spatial component uses a kernel density function.

Pros:

The model design is reasonable and it performs well on multiple synthetic and real datasets, compared against several appropriate baseline models.

The paper presentation is fairly clear, with thoroughly documented appendices.

Cons:

It is a lack of novelty. Precisely, I don’t mean the model is not novel at all; it is indeed new. But its novelty and significance is not enough.

The key design is to factor $\lambda(s, t)$ as $f(s|t) \lambda(t)$, but this design is really not well-motivated. In general, coupling $s$ and $t$ inside $\lambda$ would definitely increase the model expressiveness and, for a neural model, it often won’t increase the number of parameters. E.g., one might simply move $s$ into the power of exp in eqn-5 (and possibly allow $t$ and $s$ to interact with each other by feeding $h$ and $s$ through a neural net)---why and in what sense is the proposed factorization better than (say) this simple one?

I am sure there must be reasons to favor the proposed design, but they need to be carefully thought through and clearly presented---that is missing in the current paper.

The regularization for $w$ (suppressing the largest past $w$) is a little hacky: if one would like to balance between past and background points, considering their weights sum to 1, the most straightforward way would be to smooth the $w$ distribution, namely aggregating past $w$ and background $w$ and then increasing the entropy of the resulting binary distribution.

There are things the authors need to clarify:
(1) citations for spatiotemporal Hawkes and self-correcting processes;
(2) $\lambda$ of eqn-5 is inherited from Du et al 2016 and the authors didn’t mention it.

It is a little too bold to claim as "the first neural point process model that can jointly predict both the space and time of events": many past models can, they are just not great (e.g., they may need discretization). So I suggest the authors revise their claim: if they’d like to be a little more specific (e.g., continuous space), then their claim might be more convincing.

Questions:

The list of baseline models has Mei & Eisner 2017 but I didn’t find this model in any presented table or figure. Is this a typo or is it missing? (The authors are not obligated to compare to every single published model but they should keep the paper consistent.)

How could multiple types of events be handled in this framework? The authors use the parametrization of Du et al where they use Du’s event-type-marker to handle the event-location---this is a pity: the authors propose a new model to handle some new features, but lose the capacity of handling old features. How could they do both? E.g., in the Taxi data, in principle, one might want to model (time, location, pickup/dropoff), but the current model only handles (time, location).

---

> ### Author Response · Authors · 2020-11-24
> **Response to Reviewer 3: Expanded Justification and New Ablative Studies**
>
> Thank you for your review.
>
> __This design is really not well-motivated...Why not move s into the power of exp?__
>
> We did not use this because it suffers from numerical difficulty.  We revised our paper and included several alternative model designs and results that we have considered in Appendix D. Our design ensures a closed-form marginal intensity by summating outside the exp, such that the integration sum rule works. In contrast, moving $s$ into the power makes the marginal intensity $\lambda(t)$ intractable. When $s$ is inside exp, either learning or inference requires at least triple numerical integration, which is computationally prohibitive.
>
> __Reasons to favor the proposed design is missing in the current paper.__
>
> Thank you for the suggestion. In the revised paper, we have added more alternative designs and presented both theoretical explanations and experiment results in Appendix D to support our choice of the current design.
>
> __The regularization for $w$ is a little hacky. How about entropy regularization?__
>
> Thank you for the suggestion. We have added comparison with different regularizations, described in Section 4.3. The results are in Section 5.3 Figure 6. However, we see no significant difference in these regularizations.
>
> __The list of baseline models has Mei & Eisner 2017 but I didn’t find this model in any presented table or figure.__
>
> We did not manage to obtain good results for Mei & Eisner 2017. We did not include the results in our main text. The results are available in Appendix C2 Table 7.
>
> __The authors use the parametrization of Du et al where they use Du's event-type-marker to handle the event-location...this is a pity.__
>
> There is a misunderstanding. We did not use an event-type marker to handle location. The event-type marker is discrete.  We predict the continuous location, not discrete location.  Our framework deals with continuous space and time, not categorical features such as event-type marker.
>
> __How could multiple types of events be handled in this framework?__
>
> Our framework can be easily extended to make use of event-type markers by including the embedding of the event-type markers as part of the RNN input. This allows sharing of hidden states for multiple types of events, which can deal with sparsity of events of different types. We leave this as a future work as this is not the main focus of this paper.

---

### Official Review · AnonReviewer4 · 2020-10-28

**Rating:** 5
**Confidence:** 4

**Review:**

### Summary
This paper introduces a mechanism to learn spatio-temporal point processes using RNNs (for time estimation) and Kernel density estimation for spatial dependencies. They perform experiments on synthetic data generated by famous point processes such as Hawkes process.

### Strong/Weak points
- The paper is well-written (with a few exceptions)
- It is very easy to read and understand
- The model is very easy to understand, but very hard to reason about and to give guarantees
- It is only tested on synthetic data and no real-world example is tested upon

I have a tendency to reject this paper, as the idea is straightforward: if one is asked to mix RNNs with spatial data, the way to combine these two is very clear. There is nothing new about the theory of point processes either, hence, no new understanding of these types of processes is provided.

---

> ### Author Response · Authors · 2020-11-24
> **Response to Reviewer 4: Misunderstanding of our paper**
>
> Thank you for your review.
>
> __The model is a straightforward way to mix RNNs with spatial data, hence no novelty.__
>
> We disagree. RNNs are commonly used for evenly sampled time series data. It is known that irregularly sampled data poses significant difficulty in RNN’s learning. Our model uses a continuous-time point process to handle irregularly sampled events. The RNN component is introduced to learn the discrete jumps of the intensity function after each event. This is essentially different from a discrete-time RNN with spatiotemporal input.
>
> __The model is not tested upon real-world datasets.__
>
> We tested and provided results on three different real-world datasets, including Foursquare, NYC Taxi, and Earthquake datasets. Please see Figure 4, 5, 6, 7, and Table 7 for the experimental results using such real-world datasets.

---

### Official Review · AnonReviewer2 · 2020-10-28

**Rating:** 8
**Confidence:** 4

**Review:**

This paper sets up a spatiotemporal neural network, claiming it hasn't been done before. It may however be similar to a spatiotemporal graph network (using a regular lattice graph connectivity structure). The authors should clarify this.
The paper is well written however and sound otherwise.
Some edits are required:
- Page 2, ln 1: 'hidden states'? Language needs to be improved here.
- Section 2, ln 6: Zhao et al (2015) rather.
- Section 3 refers to 'markers'. In spatial statistics these are marks. I suggest sticking to the terminology of the field?
- Page 2, ln -4: cannot rather than can not
- Where is Figure 1 referred to in the text?
- Page 3, ln -4: What is 'firing'?
- Section 4.1, ln 1: fix the language
- Section 4.1: be consisten: 2(a) or 2 (a) - not both
- Page 6, lns -7, -8: spaces are missing
- Caption of Figure 4 needs editing.
- Figure 5: remove the names from the graphics at the top - put into captions.
- There are many capitals missing in the references and journal names that should be in full.
- Park (2019) has an et al in it?
- Is there a published version of Mozer (2017)?

---

> ### Author Response · Authors · 2020-11-24
> **Thank you and Grammar issues fixed**
>
> Thank you, we are glad that you like this paper.
>
> __The author should clarify the similarity between the model and a spatiotemporal graph network.__
>
> Our model is orthogonal to a spatiotemporal graph network. Spatiotemporal graph networks assume signals occur at regular time intervals on a discretized space (graph). Spatiotemporal point processes model events happening at irregular time intervals on a continuous space. It is also possible to model spatiotemporal point processes on a spatiotemporal graph network (think about epidemic spreading over a contact network). We believe this will be an interesting future direction.
>
> __Where is Figure 1 referred to?__
>
> Figure 1 is a visualization of the STPP models in Section 3.2. Reference added.

---

### Author Response · Authors · 2020-11-24
**Summary of the reviews and New Revision**

We thank the reviewers for their thorough and constructive feedback. We are pleased to see that they found the proposed model reasonable (R1, 3) and sound (R2) and generally acknowledged the clarity of the presentation (R1, 2, 3) of the idea. We are glad that they found our approach to be evaluated adequately on both real-world and synthetic datasets (R1, 3), though some specific questions related to experiment setup were raised, which we answer below.

One primary concern was that the design of the model is not novel enough, because we inherit the temporal component from Du et al 2016 (R1, 3). However, our main goal is not a new temporal component but to fill the gap of spatial modeling,  hence establishing a neural point process framework for spatiotemporal forecasting.

Another concern from R1 and R3 was there is still space for improvement of the model’s expressibility. However, our model design was not made “hastily”. For the revision we have
- Added more expressive models by conditioning the spatial PDF on all time steps (current and past)  (Section 4.3 and 5.3 Figure 5).
- Added ablative studies with regularizations (Section 4.3 and 5.3 Figure 6)
- Added detailed justification of our model designs and many alternatives that we have considered in Appendix D.

Through extensive experimentation, we landed with the current model design which balances the computational complexity and predictive performance.

**The edits in the updated version are in blue.**

---

### Decision · Program_Chairs · 2021-01-07
**Final Decision**

**Decision:**

Reject

**Comment:**

There was a consensus among all the reviewers that the methodological contribution is not significant enough for publication at ICLR. In short, the main contribution of the paper is to include spatial modeling into a deep temporal point process model. However, to do that, they just use a well-known method (KDE) on top of a methodology that is very similar to Du et al., KDD 2016.

In addition, in the original submission, the specific functional form for KDE was independent on time, as highlighted by the reviewers, which basically separates spatial and temporal modeling. Unfortunately, further experiments performed by the authors failed to show performance benefits on making it dependent on time.

The authors also claim that an additional contribution is the sampling method, however, this seems to thin of a contribution for a full paper.